# CD69 prevents PLZF[hi] innate precursors from prematurely exiting the thymus and aborting NKT2 cell differentiation

Motoko Y. Kimura [1,2], Akemi Igi[1], Koji Hayashizaki[1], Yukiyoshi Mita[1], Miho Shinzawa[3], Tejas Kadakia[3], Yukihiro Endo[1], Satomi Ogawa[1], Ryoji Yagi[1], Shinichiro Motohashi[2], Alfred Singer[3] & Toshinori Nakayama[1]

While CD69 may regulate thymocyte egress by inhibiting $S1P_1$ expression, CD69 expression is not thought to be required for normal thymocyte development. Here we show that CD69 is in fact specifically required for the differentiation of mature NKT2 cells, which do not themselves express CD69. Mechanistically, CD69 expression is required on CD24[+] PLZF[hi] innate precursors for their retention in the thymus and completion of their differentiation into mature NKT2 cells. By contrast, CD69-deficient CD24[+] PLZF[hi] innate precursors express $S1P_1$ and prematurely exit the thymus, while $S1P_1$ inhibitor treatment of CD69-deficient mice retains CD24[+] PLZF[hi] innate precursors in the thymus and restores NKT2 cell differentiation. Thus, CD69 prevents $S1P_1$ expression on CD24[+] PLZF[hi] innate precursor cells from aborting NKT2 differentiation in the thymus. This study reveals the importance of CD69 to prolong the thymic residency time of developing immature precursors for proper differentiation of a T cell subset.

[1] Department of Immunology, Graduate School of Medicine, Chiba University, 1-8-1 Inohana, Chuo-ku, Chiba 260-8670, Japan. [2] Department of Medical Immunology, Graduate School of Medicine, Chiba University, 1-8-1 Inohana, Chuo-ku, Chiba 260-8670, Japan. [3] Experimental Immunology Branch, National Cancer Institute, National Institutes of Health, Bethesda, MD 20892, USA. Correspondence and requests for materials should be addressed to M.Y.K. (email: kimuramo@chiba-u.jp)

CD4+CD8+ double-positive (DP) thymocytes are the first cells to express rearranged αβ T cell receptors (TCRs) and to be signaled to differentiate into conventional CD4 and CD8 T cells, regulatory T cells or invariant natural killer T (iNKT) cells. Through mechanisms that are still not fully understood, the differentiation of DP thymocytes into these different cell fates is determined by the specificity of their TCRs for different types of selecting ligands in the thymus[1]. TCRs that recognize self-peptides presented by MHC complexes on cortical thymic epithelial cells signal differentiation into conventional CD4 or CD8 mature naive T cells, and these mature naive T cells do not acquire an effector function until after their encounter with agonist ligands in the periphery. In contrast, TCRs that recognize glycolipid antigens presented by cortical thymocytes signal differentiation into three distinct subsets of iNKT effector cells: NKT1, NKT2, and NKT17 cells[2–6], which can be signaled to rapidly produce interferon-γ (IFNγ), interleukin-4 (IL-4), and IL-17 effector cytokines, respectively, during their development in the thymus. These three iNKT subsets are thought to arise as completely distinct effector subsets from iNKT precursors in the thymus[7], but it is not known how iNKT precursors in the thymus are signaled to adopt different iNKT effector lineage fates. Furthermore, it has been suggested that each iNKT subset has a distinct function, in which the IFNγ production by iNKT cells (i.e., NKT1 cells) is crucial for anti-microbial and anti-tumor immunity[2,8], whereas the IL-4 production by iNKT cells (i.e., NKT2 cells) in early virus infection is crucial for germinal center formation and anti-viral antibody production[9].

In the thymus, TCR signaling of developing thymocytes induces the expression of CD69, a type 2 transmembrane protein with a C-type lectin-like domain that is encoded within the NK gene cluster on chromosome 6 in mice and chromosome 12 in humans[10,11]. Importantly, CD69 directly competes with $S1P_1$, a chemokine receptor that is required for thymocyte egress, for surface expression on thymocytes[12–14]. As a result, thymocytes that have completed their differentiation must down-regulate CD69 surface expression in order to express surface $S1P_1$ so that they can leave the thymus and emigrate into the periphery. Consequently, CD69 expression might be important for preventing immature thymocytes from prematurely exiting the thymus so that they can be retained until their differentiation is complete. However, this perspective has never been experimentally validated, as studies with genetically CD69-deficient ($Cd69^{-/-}$) mice have not identified problems in the development of conventional CD4 or CD8 T cells[15,16]. In fact, no differentiation pathway has yet been shown to require CD69-mediated thymic retention or CD69-mediated prolongation of thymic residency, although the role of CD69-mediated thymic retention in iNKT cell development has not been assessed.

In the present study, we find that CD69 expression on CD24+ PLZF$^{hi}$ innate precursors is crucial for the in vivo differentiation of mature NKT2 cells. Endogenous CD69 expression on CD24+ PLZF$^{hi}$ innate precursors is necessary for their retention in the thymus so that they can differentiate into mature NKT2 cells. Indeed, the strict requirement for CD69 expression on CD24+ PLZF$^{hi}$ innate precursors can be replaced by specific inhibitors of $S1P_1$ signaling that prolong thymic residency time and allow CD24+ PLZF$^{hi}$ innate precursors to differentiate into mature NKT2 cells. This study documents the importance of the CD69-mediated prolongation of the thymic residency time for the differentiation of a T cell subset and enhances the understanding of when and how distinct effector fates are determined during iNKT cell development in the thymus.

## Results

**Expression of NK cluster genes in each iNKT subset.** iNKT cells are unique T cells that express the molecule NK1[17,18], which is encoded by *Klrb1c*, a gene within the NK gene cluster on murine chromosome 6 (Fig. 1a, left). We began the present study by exploring whether iNKT cells also express other NK cluster genes in addition to NK1 and if their pattern of expression varies in different NKT subsets. To do so, we purified mature iNKT cell subsets from the thymus of BALB/c-background mice (Supplementary Fig. 1a, b) and determined their expression of NK cluster genes (Fig. 1a, right). To our surprise, each mature iNKT subset displayed a different NK gene expression pattern: NKT1 cells expressed all NK cluster genes; NKT2 cells contained little expression of any NK cluster genes, including CD69; and NKT17 cells expressed *Cd69* and all genes encoded upstream of *Cd69* (i.e., *Klrb1c* and *Klrb1f*) but no genes encoded downstream of *Cd69* (i.e., *Klre1*, *Klrd1*, *Klrk1*, *Klrc2*, *Klrc1*, and *Klra1*) in the NK gene cluster (Fig. 1a, right). We also assessed the NK gene expression at the protein level by examining the surface expression of three proteins: CD94, which is encoded by *Klrd1*; NKG2D, which is encoded by *Klrk1*; and CD69, which is encoded by *Cd69*. Similar to their expression of NK gene messenger RNAs (mRNAs), NKT1 cells expressed high surface amounts of all three proteins (CD94, NKG2D, CD69); NKT2 cells expressed little or none of these proteins; and NKT17 cells expressed surface CD69 but neither CD94 nor NKG2D, which are both encoded downstream of *Cd69* in the NK gene cluster (Fig. 1b, c, Supplementary Fig. 1c, Supplementary Fig. 1d, and Supplementary Fig. 1e). Thus, each mature iNKT cell subset (NKT1, NKT2, NKT17) displayed a unique expression pattern of NK cluster genes[19,20].

Because the NK cluster gene expression differed in each mature iNKT subset, we assessed the NK cluster gene expression earlier in iNKT cell development and prior to differentiation into mature iNKT effector subsets. In these experiments, we identified newly arising CD1d.PBS57 tetramer-binding iNKT cells in the thymus of C57BL/6-background mice expressing the $Rag2^{GFP}$ transgene. Since immature CD24+ iNKT cells are known to be small, non-cycling cells, unlike mature CD24– iNKT cells, which are large proliferating cells[21], we were able to utilize the linear decay of the Rag2-GFP protein to confirm the temporal sequence of immature CD24+ iNKT cell development in the thymus[22,23]. Newly arising tetramer-binding cells are CD24+ thymocytes that have classically been identified as stage 0 cells that differentiate into CD24– stage 1 cells[24]. However, we found that stage 0 iNKT cells actually consist of two developmentally distinct CD44$^{lo}$ and CD44$^{hi}$ subpopulations, which we termed stage 0a and stage 0b, respectively (Fig. 2a). Phenotypically, the stage 0a (CD24+CD44$^{lo}$) cells were CD4+CD8+ and expressed Rag2-GFP at high levels that were only slightly lower than those in unsignaled DP thymocytes (Fig. 2a, Supplementary Fig. 2a, and Supplementary Fig. 2b), indicating that stage 0a cells had very recently been TCR signaled. Stage 0b (CD24+CD44$^{hi}$) cells were also CD4+CD8+ but expressed Rag2-GFP at lower levels, indicating that stage 0b cells developed after stage 0a cells (Fig. 2a, Supplementary Fig. 2a, and Supplementary Fig. 2b). Stage 1 (CD24–CD44$^{lo}$) cells were no longer DP but had phenotypically converted to either CD4SP or double-negative thymocytes and expressed distinctly lower levels of Rag2-GFP than stage 0 cells (Fig. 2a, Supplementary Fig. 2a, and Supplementary Fig. 2b). Stage 2 and 3 (CD24–CD44$^{hi}$) cells no longer expressed Rag2-GFP (Fig. 2a, Supplementary Fig. 2a, and Supplementary Fig. 2b) and so were even later appearing, consistent with their containing mature iNKT effector subsets. These results reveal that stage 0 subsets (stage 0a and stage 0b) and stage 1 cells are "early" stages of iNKT cells that still express Rag2-GFP. Importantly, based on the CD44 expression, we are

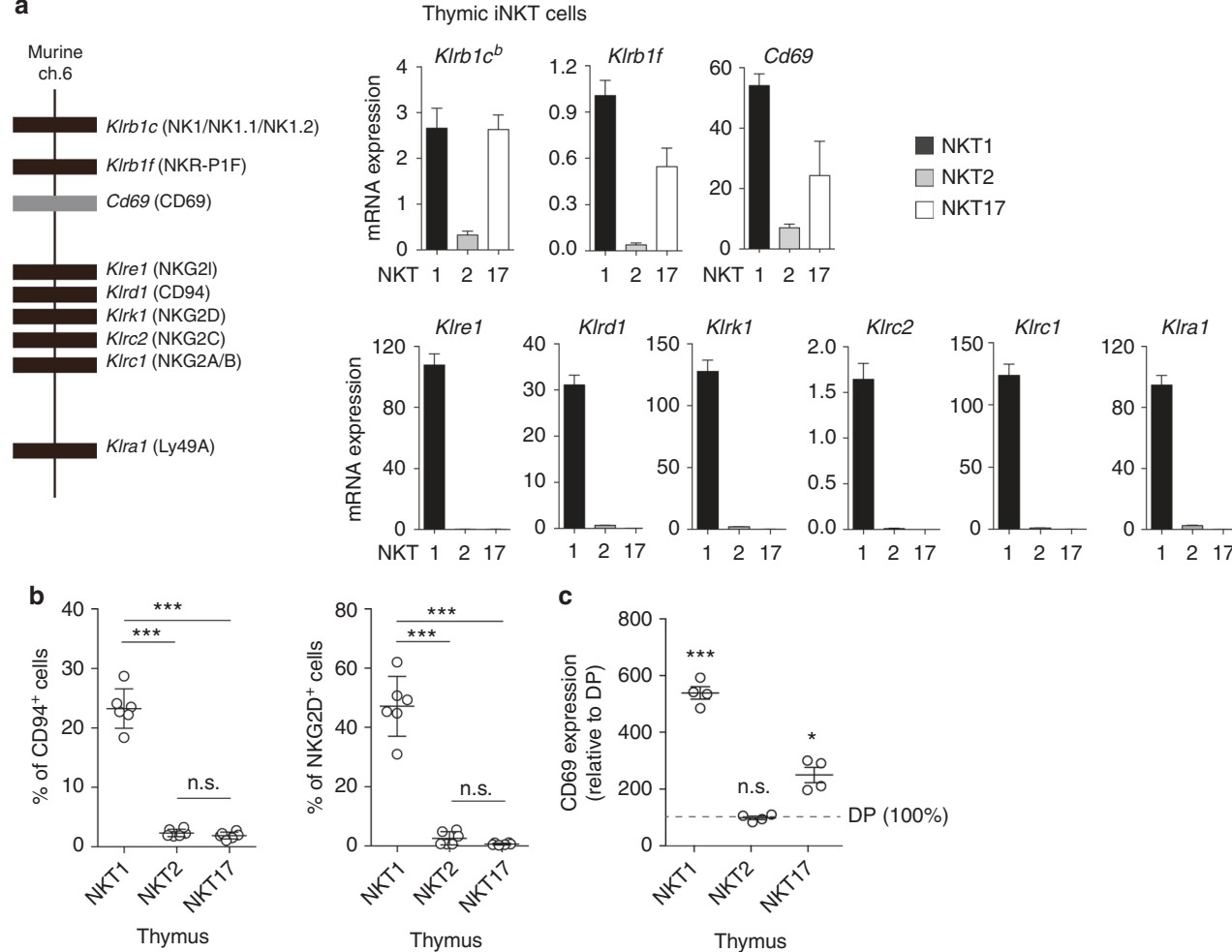

**Fig. 1** NK cluster gene expression in each iNKT subset. **a** Murine NK cluster genes, including the *Cd69* gene, are located on murine chromosome 6 (left). The mRNA expression of NK cluster genes in NKT1 (CD3$^+$ CD1d.PBS57$^{low}$ CD138$^-$ CD44$^{hi}$), NKT2 (CD3$^+$ CD1d.PBS57$^{hi}$ CD138$^-$) and NKT17 (CD3$^+$ CD1d. PBS57$^+$ CD138$^+$) cells sorted from BALB/c thymus and analyzed by quantitative RT-PCR. Data are shown relative to the *Rpl13a* expression (right). **b** A flow cytometry analysis of the frequencies of CD94$^+$ or NKG2D$^+$ cells gated on each iNKT subset (identified as in Supplementary Fig. 1c) from BALB/c thymus. **c** The CD69 expression on each iNKT subset (identified as in Supplementary Fig. 1c) from BALB/c thymus presented as the mean fluorescence intensity relative to that on preselection (TCRβ$^{lo-med}$ CD4$^+$CD8$^+$) thymocytes. The mean and SEM are shown. *$p < 0.05$ and ***$p < 0.001$ (one-way ANOVA). Representative data from five experiments with five mice are shown (**a**). Data are pooled from two experiments with six mice (**b**) and three experiments with four mice (**c**). n.s. not significant

now able to divide stage 0 iNKT cells into earlier and later stage 0 subsets (stage 0a and stage 0b) in order to better study the very-early events of iNKT cell development.

Having determined the three different immature iNKT cells (i.e., stage 0a, stage 0b, and stage 1), we then assessed their surface expression of CD69, CD94, and NKG2D proteins encoded in the NK gene cluster. CD94 and NKG2D were not expressed on stage 0a or stage 1 immature iNKT cells, but were expressed on a few stage 0b cells and increased their expression during terminal differentiation into mature NKT1 cells (Figs. 1b, 2b and Supplementary Fig. 2c), indicating that expression of CD94 and NKG2D was unlikely to be important for early iNKT cell development. In contrast, CD69 was expressed in all early-developing iNKT cells (stage 0a, stage 0b, and stage 1), with its highest expression on stage 0b cells, suggesting that the CD69 expression might be important in early iNKT cell development (Figs. 1c, 2c and Supplementary Fig. 2d).

**Impact of CD69 deficiency on NKT2 cell differentiation.** To determine if CD69 plays an important role in early iNKT cell development, we examined the iNKT cell development in the thymus of *Cd69*$^{-/-}$ BALB/c-background mice with germline deletion of the *Cd69* gene. We found that CD69 deficiency significantly reduced the overall frequency of iNKT cells in the thymus, and, more importantly, markedly altered the effector lineage direction of iNKT thymocyte differentiation. Indeed, the percentage of iNKT cells that were mature NKT1 cells was significantly higher and the percentage that were mature NKT2 cells significantly lower in the thymus of *Cd69*$^{-/-}$ BALB/c-background mice than in that of *Cd69*$^{+/+}$ mice, while the relative percentage that were mature NKT17 cells was unchanged (Fig. 3a). The findings were similar when the NKT1 and NKT2 cell frequencies were expressed as a percentage of total thymocytes (Supplementary Fig. 3a, upper). We also confirmed that the absolute numbers of NKT1 and NKT2 cells in the thymus showed the same results (Supplementary Fig. 3a, bottom). Thus, the presence of CD69 expression markedly affected the differentiation of thymocyte precursors into mature NKT1 and/or NKT2 cells.

A time-course study from birth to 3 months of age revealed that CD69 deficiency altered NKT1 and NKT2 cell subsets from

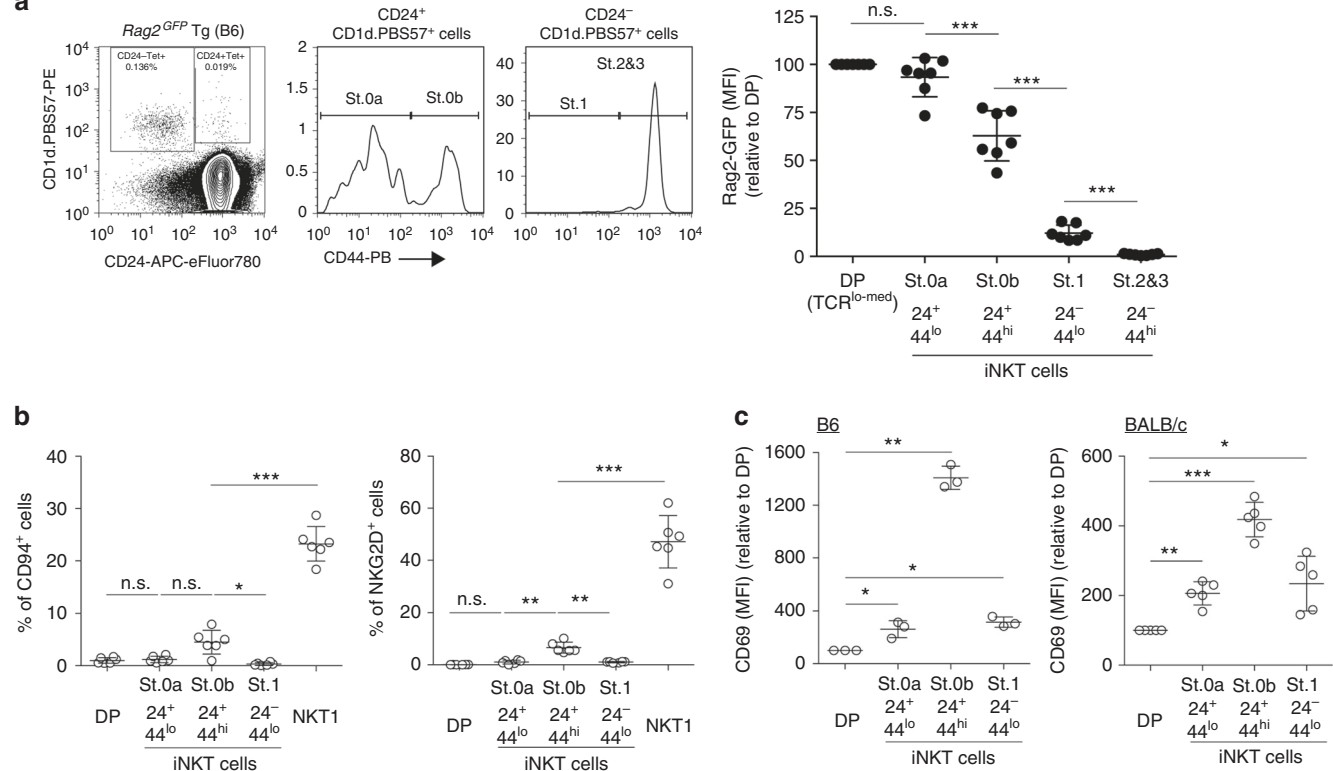

**Fig. 2** CD24[+]NKT precursors express CD69 but not other NK cluster genes. **a** The identification of cells early after positive selection using $Rag2^{GFP}$ Tg mice (C57BL/6-background). CD24[+]CD44[lo] cells are the earliest iNKT cell progenitors (stage 0a, St. 0a) with a high RAG2-GFP expression, CD24[+]CD44[med/hi] cells have the second highest RAG2-GFP expression (stage 0b, St. 0b), CD24[-]CD44[low] cells have low Rag2-GFP expression (stage 1, St. 1), and CD24[-]CD44[hi] cells have no Rag2-GFP expression (stages 2 and 3, St. 2 and 3). The RAG2-GFP levels in preselection thymocytes (TCRβ[lo-med] CD4[+]CD8[+]) were set at 100, and iNKT precursors are shown relative to preselection thymocytes. **b** Frequencies of CD94[+] or NKG2D[+] cells in each subset of iNKT cells from BALB/c thymus. **c** The CD69 expression on preselection DP cells (DP), stage 0a (St. 0a), stage 0b (St. 0b), and stage 1 (St. 1) of iNKT cells from C57BL/6 (left) and BALB/c (right) thymus presented relative to preselection thymocytes. The mean and SEM are shown. *$p < 0.05$, **$p < 0.01$, and ***$p < 0.001$ (one-way ANOVA). Representative data from four experiments with seven mice are shown (**a**). Data are pooled from two experiments with six mice (**b**) and three experiments with three (C57BL/6) or six mice (BALB/c) (**c**). n.s. not significant

the earliest time that iNKT cells appeared (at 1 week of age) (Fig. 3b). Few iNKT cells were detected in either $Cd69^{+/+}$ or $Cd69^{-/-}$ mice before 2 weeks of age, after which the timepoint iNKT cell frequency sharply increased in both strains (Fig. 3b and Supplementary Fig. 3b). Notably, the percentage of the NKT1 subset among iNKT cells was markedly higher and the percentage of the NKT2 subset markedly lower in $Cd69^{-/-}$ mice than in $Cd69^{+/+}$ mice, whereas the percentage of the NKT17 subset remained roughly the same between the two strains (Fig. 3b and Supplementary Fig. 3b). Importantly, the percentages of NKT2 cells and the absolute number of NKT2 cells in the spleen and lung of $Cd69^{-/-}$ mice were decreased, results that are similar to those in the thymus (Fig. 3c and Supplementary Fig. 3c), indicating that CD69 deficiency did indeed alter their differentiation in the thymus.

The assessment of C57BL/6-background mice with fewer iNKT cells than BALB/c-background mice revealed that CD69 deficiency significantly reduced the NKT2 subset, but did not alter the NKT1 subset in $Cd69^{-/-}$ C57BL/6-background mice (Supplementary Fig. 3d). Consequently, we think that the primary impact of CD69 deficiency is the impaired differentiation of precursor thymocytes into the mature NKT2 subset and that increases in the NKT1 subset frequencies are secondary to the reduced generation of NKT2 cells. This is especially prominent in BALB/c-background mice because of their high iNKT cell number. To examine whether this specific reduction of NKT2 cell differentiation in the absence of CD69 was cell intrinsic, we

created a mixed BM chimera, in which the bone marrow (BM) cells from either $Cd69^{+/+}$ or $Cd69^{-/-}$ BALB/c-background mice mixed together with BM cells from $J\alpha18^{-/-}$ BALB/c-background mice were transferred into lethally irradiated $J\alpha18^{-/-}$ BALB/c-background host mice. In these chimeric mice, all iNKT cells should have been derived from either $Cd69^{+/+}$ or $Cd69^{-/-}$ BM cells and developed under the environment where CD69-sufficient cells exist (Supplementary Fig. 3e). We found that the generation of NKT2 cells from $Cd69^{-/-}$ BM cells was significantly decreased, whereas the NKT1 and NKT17 cell generation was not affected (Supplementary Fig. 3e). These data demonstrate that the reduction in the number of NKT2 cells in the absence of CD69 is caused by a cell-intrinsic mechanism and not by defects in the development of other cells that might be important for NKT2 cell differentiation.

We therefore conclude that CD69 expression on early iNKT cells in the thymus is required for their differentiation into mature NKT2 cells, even though mature NKT2 cells do not themselves express CD69.

**CD69 prevents CD24[+]PLZF[hi] precursors from leaving the thymus.** To gain insight into why NKT2 cell differentiation depends on CD69 expression, we considered the fact that CD69, by interacting with and suppressing S1P$_1$ surface expression, inhibits thymocyte egress from the thymus[12–14]. Because mature NKT2 cells do not express CD69 (Fig. 1c), CD69 deficiency could

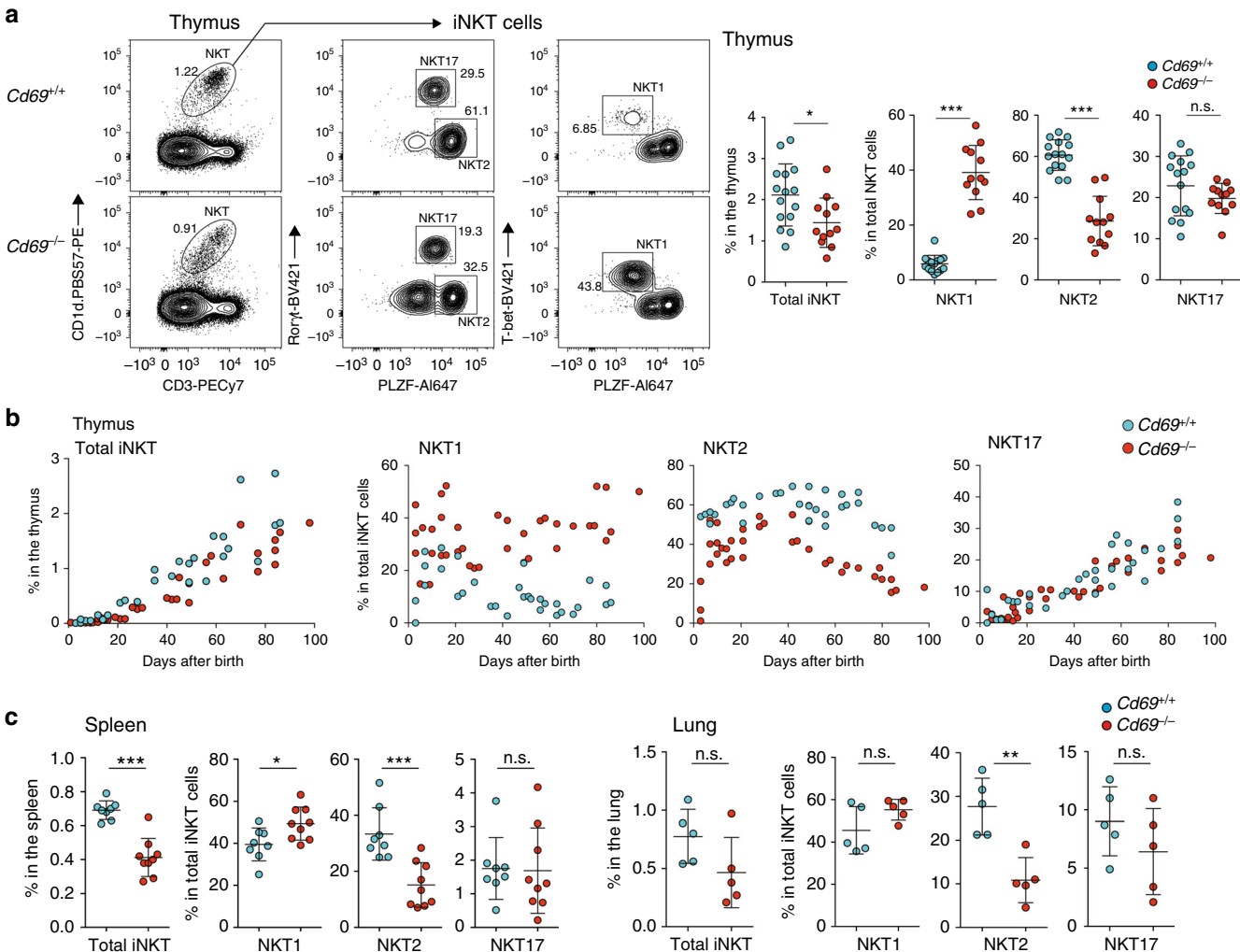

**Fig. 3** Diminished NKT2 cell differentiation in *Cd69*⁻/⁻ mice. **a**, **c** Profiles of iNKT subsets in the thymus (**a**) and frequencies of iNKT subsets in the spleen and lung (**c**) from *Cd69*⁺/⁺ and *Cd69*⁻/⁻ BALB/c-background mice. The total frequency of iNKT cells and the frequencies of NKT1 (PLZF^lo T-bet⁺), NKT2 (PLZF^hi Rorγt^med), and NKT17 (PLZF^med Rorγt^hi) cells among total iNKT cells are shown. **b** Ontogeny of iNKT subsets in BALB/c thymus, showing the total frequency of iNKT cells and the frequencies of NKT1 (PLZF^lo T-bet⁺), NKT2 (PLZF^hi Rorγt^med), and NKT17 (PLZF^med Rorγt^hi) cells among iNKT cells up to 100 days after birth. The mean and SEM are shown. *$p < 0.05$, **$p < 0.01$, and ***$p < 0.001$ (two-tailed unpaired *t* test). Representative data from 8 to 11 experiments with more than 10 mice is shown (**a**). Data are pooled from more that 10 experiments with 30 (*Cd69*⁺/⁺) or 37 (*Cd69*⁻/⁻) mice (**b**), 4-8 experiments with more than 8 mice (spleen), or 5 mice (lung) (**c**). n.s. not significant

only have affected NKT2 cells at a previous developmental step. Accordingly, we thought that *Cd69*⁻/⁻ CD24⁺ iNKT cells might have prematurely exited the thymus because, in the absence of CD69, these cells might express S1P₁. Indeed, *S1pr1* mRNA was found to be expressed in CD24⁺ iNKT precursors, and its expression was significantly lower in CD24⁺ iNKT cells from *Cd69*⁻/⁻ mice than in those from *Cd69*⁺/⁺ mice (Fig. 4a), suggesting that *Cd69*⁻/⁻ CD24⁺ iNKT cells with high *S1pr1* mRNA had exited the thymus.

We next attempted to detect CD24⁺ iNKT cells in the periphery and found that there were CD24⁺CD1d.PBS57⁺ cells in the spleen of BALB/c mice (Fig. 4b red). However, the percentage of CD24⁺CD1d.PBS57⁺ cells was extremely low, and the high background staining of CD24⁺CD1d.control⁺ cells (Fig. 4b, blue) prevented us from concluding actual existence of CD24⁺iNKT cells in the spleen. Accordingly, we next tried to identify specific markers to detect CD24⁺ iNKT cells in the spleen and found that CD24⁺CD1d.PBS57⁺ cells (red) from *Cd69*⁺/⁺ and *Cd69*⁻/⁻ mice, but not from *Jα18*⁻/⁻ mice specifically contained CD4⁺ and CD44^hi cells (Fig. 4b, right),

whereas CD24⁺CD1d.control⁺ cells (blue) neither contained CD4⁺ nor CD44^hi cells (Fig. 4b, right). These data suggest that CD4⁺CD44^hi CD24⁺CD1d.PBS57⁺ cells are real iNKT cells. We next examined if these CD4⁺CD44^hi CD24⁺CD1d.PBS57⁺ cells are truly recent thymic emigrants using *Rag2*^GFP C57BL/6 mice (Fig. 4c). Similar to the data from BALB/c-background mice (Fig. 4b), CD24⁺CD1d.PBS57⁺ cells (red) contained CD44^hi cells, but CD24⁺CD1d.cntrl⁺ cells (blue) did not (Fig. 4c, middle right), indicating that the CD44^hi CD24⁺CD1d.PBS57⁺ cells are truly iNKT cells. Importantly, some of these CD44^hi CD24⁺CD1d.PBS57⁺ cells expressed Rag2-GFP (Fig. 4c, right), demonstrating that these cells were recent thymic emigrants. Since these Rag2-GFP⁺CD44^hi CD24⁺CD1d. PBS57⁺ cells were detected even in the *Cd69*⁺/⁺ mice, we named these precursors "early-exiting precursors" (Supplementary Fig. 5). These data suggest that CD24⁺ iNKT precursors contain distinct subsets, with early-exiting precursors prematurely leaving the thymus and NKT2 precursors (but not NKT1 or NKT17 precursors) potentially able to prematurely exit the thymus in the absence of CD69 (Supplementary Fig. 5).

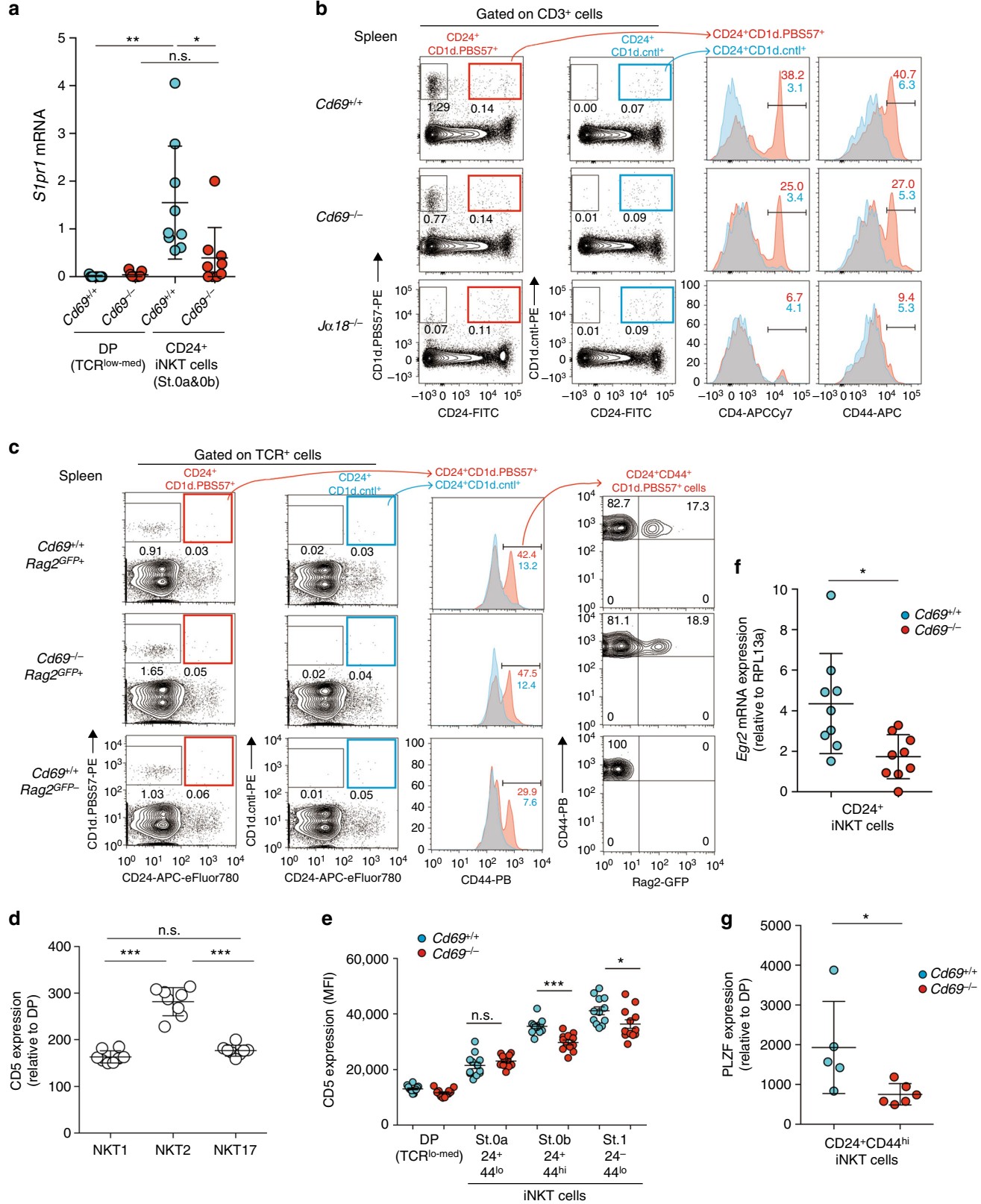

Differentiation of mature NKT2 cells has been suggested to require high-intensity TCR signaling[7,25]. In fact, mature NKT2 cells displayed the highest cell-surface CD5 of the three mature iNKT subsets (Fig. 4d), indicating that NKT2 cells had received the highest-intensity TCR signaling among the three iNKT subsets. To examine whether in the absence of CD69, NKT2 precursors preferentially exit the thymus early, we considered the possibility that high-intensity TCR signals might generate distinct NKT2 precursors (Supplementary Fig. 5). During early iNKT cell development, the expression of both CD69 and CD5 is increased

**Fig. 4** CD69 prevents CD24+ PLZF^hi precursors from leaving the thymus. **a** The $S1pr1$ mRNA expression in sorted preselection cells and CD24+ iNKT cells (stage 0a and stage 0b) from the thymus from BALB/c-background mice, presented relative to the $Rpl13a$ expression. *$p < 0.05$, and **$p < 0.01$ (two-tailed unpaired $t$ test). **b**, **c** A flow cytometry analysis of CD24 vs. CD1d.PBS57 or CD1d.control staining of CD3+ splenocytes from the indicated mice of BALB/c (**b**, left) or C57BL/6-background (**c**, left). The number indicates the percentage of gated cells. The CD4 (**b**) and CD44 (**b**, **c**) expression of splenic CD24 +CD1d.PBS57+ cells (red) or splenic CD24+CD1d.control+ cells (blue) are shown (right). The number indicates the percentage of CD4+ or CD44^hi cells gated. Profiles of RAG2-GFP vs. CD44 expression of splenic CD24+CD44^hiCD1d.PBS57+ cells are shown (**c**, right). The number indicates the percentage of gated cells. **d** A flow cytometry analysis of the CD5 expression on NKT1, NKT2, and NKT17 cells in the thymus of BALB/c-background mice, presented relative to preselection DP thymocytes. ***$p < 0.001$ (one-way ANOVA). **e** The mean fluorescence intensity of the CD5 expression in each stage of iNKT cells from BALB/c-background mice. **f** The $Egr2$ mRNA expression in sorted CD24+ iNKT cells (stage 0a and 0b cells) from BALB/c-background mice, presented relative to $Rpl13a$. **g** The PLZF expression on each stage of iNKT cells from BALB/c-background mice, presented relative to that on preselection thymocytes. *$p < 0.05$, **$p < 0.01$, and ***$p < 0.001$ (two-tailed unpaired $t$ test for **e**–**g**). The mean and SEM are shown. Data pooled from four experiments with four mice is shown (**a**), and representative data from three experiments with six ($Cd69^{+/+}$), five ($Cd69^{-/-}$), or five ($J\alpha18^{-/-}$) mice are shown (**b**). Representative data from five experiments with more than 10 ($Cd69^{+/+}$), 8 ($Cd69^{+/+}Rag2^{GFP+}$), or 7 ($Cd69^{-/-}Rag2^{GFP+}$) mice are shown (**c**). Data are pooled from more than 8 experiments with 8 mice (**d**), 8 experiments with more than 10 mice (**e**), 3 experiments with 3 mice (**f**), and 3 experiments with 5 ($Cd69^{+/+}$) or 6 ($Cd69^{-/-}$) mice (**g**). n.s. not significant

between stage 0a and stage 0b cells (Figs. 2c and 4e), whereas the CD5 surface expression on stage 0b cells was found to be significantly lower on $Cd69^{-/-}$ thymocytes than $Cd69^{+/+}$ thymocytes (Fig. 4e). These data indicate that strongly signaled NKT2 precursors with a high CD5 expression left the thymus in the absence of CD69, resulting in the low CD5 expression on the remaining stage 0b cells in the thymus.

To link TCR signaling intensity in early iNKT cells with their lineage direction, we next examined the Egr2 and PLZF expression in stage 0b cells. TCR signaling is known to quantitatively upregulate Egr2, and Egr2 induces PLZF, the iNKT lineage-specifying transcription factor[26,27]. We found that the $Egr2$ mRNA and PLZF protein expression in stage 0 $Cd69^{-/-}$ thymocytes was significantly reduced (Fig. 4f, g), indicating that CD24+ NKT2 precursors with high amounts of Egr2 and PLZF left the thymus, thereby reducing the expression of Egr2 and PLZF in the remaining stage 0b $Cd69^{-/-}$ thymocytes. We named these NKT2 precursors that left the thymus prematurely in the absence of CD69 "PLZF^hi innate precursors". We thus conclude that CD24+ PLZF^hi innate precursors express high S1P₁, CD5, Egr2, and PLZF as a result of strong TCR signaling, and these PLZF^hi innate precursors prematurely leave the thymus in the $Cd69^{-/-}$ mice, resulting in the failure of these precursors to become mature NKT2 cells (Supplementary Fig. 5).

**Loss of S1P₁ expression rescues NKT2 cell differentiation**. Since CD69 prevents the cell-surface expression of S1P₁, which signals thymic egress, we next examined whether S1P₁ deficiency in $Cd69^{-/-}$ mice prevented early thymic egress of CD24+ PLZF^hi innate precursors and restored NKT2 cell differentiation in the $Cd69^{-/-}$ thymus. To do so, we crossed $Cd69^{-/-}$ mice with $S1pr1^{f/f}$ $Cd4$-Cre transgenic (Tg) mice (hereafter known as "$Cd69^{-/-}$ $S1pr1^{-/-}$" mice). In contrast to its significant effect of S1P₁ deficiency on retaining CD4SP and CD8SP cells in the $Cd69^{-/-}$ thymus (Supplementary Fig. 6a)[12,13], S1P₁ deficiency only slightly increased the total frequency of CD69-deficient iNKT cells in the thymus (Fig. 5a and Supplementary Fig. 6a). Notably, the increased frequency of iNKT cells was mainly due to NKT2 cells, since the NKT2 cell frequency in the thymus was most significantly increased (Fig. 5b). Furthermore, the NKT2 cell frequency was significantly increased in the thymus of $Cd69$ Tg mice, in which the S1P₁ surface expression was completely prevented[28] (Supplementary Fig. 6b). These data suggest that S1P₁ deficiency preferentially impacts NKT2 cells.

In addition, the CD5 expression on CD69-deficient CD24 +CD44^hi (stage 0b) iNKT cells was significantly increased in $S1pr1^{-/-}$ compared to $S1pr1^{+/+}$ mice (Fig. 5c), indicating that S1P₁ deficiency prevents CD69-deficient CD5^hi PLZF^hi innate

precursors from leaving the thymus, and, consequently, the overall CD5 expression on stage 0b cells is increased.

In the absence of S1P₁, NKT2 cell differentiation is rescued in $Cd69^{-/-}$ mice. Of note, the NKT2 cell differentiation was significantly increased in S1P₁-deficient cells even in CD69-sufficient cells, and the CD5 expression on S1P₁-deficient stage 0b cells was also increased in CD69-sufficient cells (Fig. 5b, c). These data suggest that S1P₁ deficiency prevents CD69-sufficient CD24+ iNKT precursors, which we named "early-exiting precursors" (Fig. 4b, c and Supplementary Fig. 5), from leaving the thymus. Since these early-exiting precursors have a high CD5 expression, they must have received stronger TCR signaling, which promotes differentiation to mature NKT2 cells when they remain in the thymus.

Since $S1pr1^{-/-}$ mice are on a C57BL/6 background, the frequency of NKT2 cells is much lower in these mice than in those with a BALB/c background (Supplementary Fig. 3d). Thus, we next examined the impact of S1P₁ inhibition on the iNKT cell differentiation in BALB/c-background mice using FTY720, an S1P₁ agonist that induces S1P₁ internalization. In vivo injection for five consecutive days of FTY720 slightly but significantly increased the total frequency of iNKT cells in the thymus (Fig. 5d and Supplementary Fig. 6c), and this increase was specifically a consequence of the increased number of NKT2 cells (Fig. 5e and Supplementary Fig. 6d). In contrast, the significant inhibition of the thymic egress of both CD4SP and CD8SP cells from the thymus was detected[12] (Supplementary Fig. 6e), suggesting that FTY720 treatment was effective. Notably, FTY720 treatment significantly increased the frequency of NKT2 cells and decreased that of NKT1 cells (Fig. 5e), demonstrating that diminished NKT2 cell generation in $Cd69^{-/-}$ mice is substantially rescued by FTY720 treatment. Furthermore, the CD5 expression in CD24 +CD44^hi (stage 0b) iNKT precursors was increased by FTY720 treatment (Fig. 5f), demonstrating that FTY720 treatment prevents CD5^hi PLZF^hi innate precursors from leaving the thymus.

We therefore conclude that CD69 is required to prevent CD24 +CD5^hiPLZF^hi innate precursors from leaving the thymus by inhibiting the S1P₁ expression, resulting in their differentiation into mature NKT2 cells in the thymus (Supplementary Fig. 5).

**Impact of let-7 miRNA on the PLZF expression in stage 0b cells**. Our results indicate that CD24+CD44^hi (stage 0b) cells contain multiple different precursor subsets: CD5^hiPLZF^hi NKT2 precursors, CD5^lowPLZF^low NKT1 precursors and early-exiting precursors (Supplementary Fig. 5). Because the PLZF expression in iNKT cells is known to be directly regulated by let-7 miR-NAs[29], we investigated whether stage 0b cells that received weak

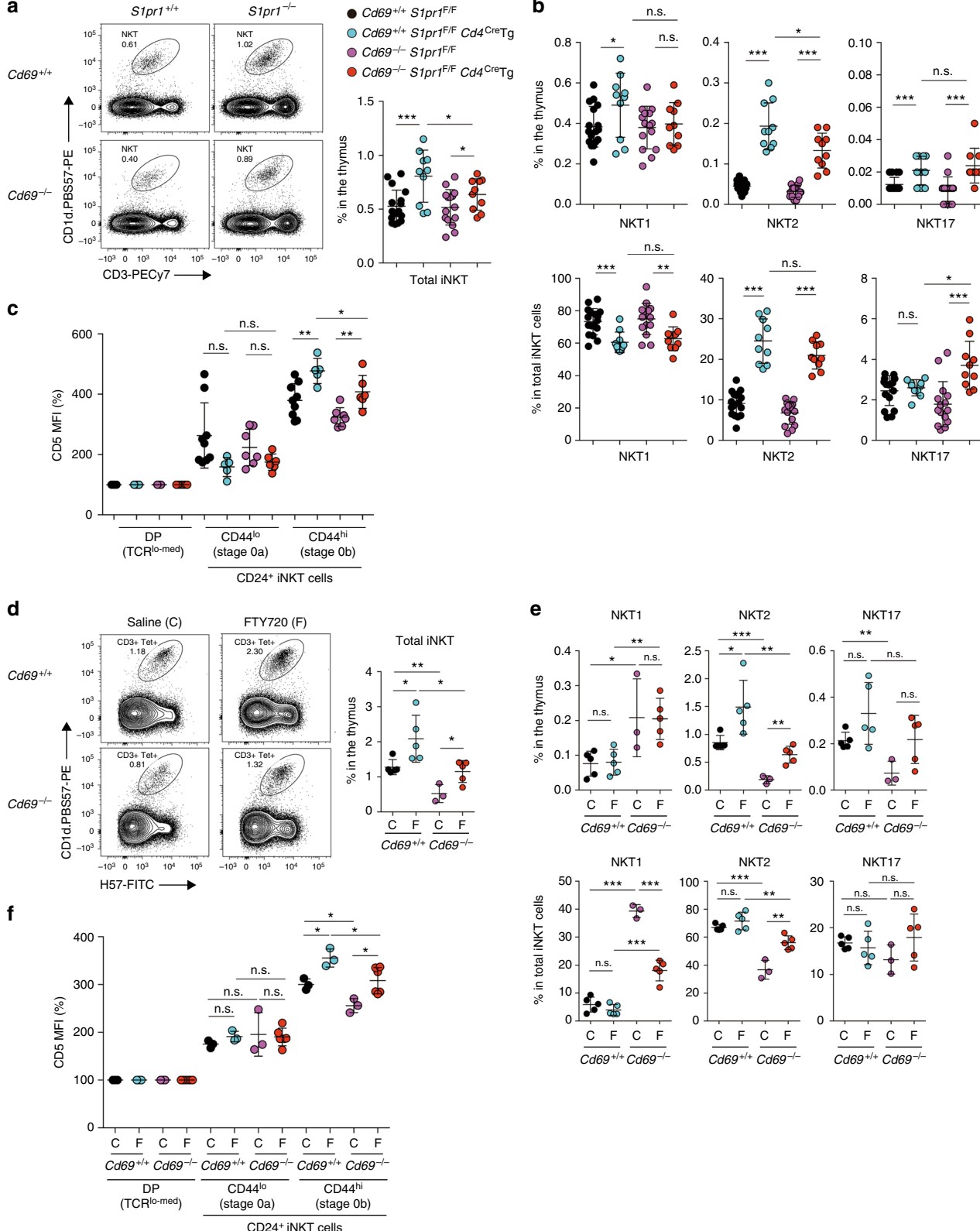

TCR signaling had a low expression of PLZF, as PLZF was targeted by let-7 miRNAs.

To determine if the absence of let-7 miRNA increased the PLZF expression in stage 0b cells and subsequently increased NKT2 cell differentiation, we introduced Lin28A Tg, which inhibits let-7 miRNA expression[29,30]. We found that Lin28A Tg dramatically induced the PLZF expression in CD24+ (stage 0) precursors, even in CD69-deficient C57BL/6-background mice (Fig. 6a and Supplementary Fig. 7), showing that stage 0b precursors have high PLZF levels when let-7 miRNA expression is prevented. We interpret these data as revealing that stage 0b precursors that have received weak TCR signaling contain let-7

**Fig. 5** Prevention of S1P$_1$ surface expression rescues NKT2 cell differentiation. **a** Profiles of iNKT cells in the thymus from $Cd69^{+/+}$, $Cd69^{-/-}$, $S1pr1^{-/-}$, $Cd69^{-/-}S1pr1^{-/-}$ C57BL/6-background mice (left). The total frequencies of iNKT cells in the thymus are shown (right). **b** The frequencies of NKT1 (PLZF$^{lo}$T-bet$^+$), NKT2 (PLZF$^{hi}$Rorγt$^{med}$), and NKT17 (PLZF$^{med}$Rorγt$^{hi}$) cells in the thymus (upper) and among total iNKT cells (bottom) are shown. **c** The CD5 expression on each stage of cells presented relative to that on preselection thymocytes (DP). **d** Profiles of iNKT cells in the thymus from $Cd69^{+/+}$ and $Cd69^{-/-}$ BALB/c-background mice after 5 consecutive daily injections with either saline (**C**) or the S1P$_1$ agonist FTY720 (**F**). The total frequency of iNKT cells in the thymus is shown. **e** The frequencies of NKT1 (PLZF$^{lo}$T-bet$^+$), NKT2 (PLZF$^{hi}$Rorγt$^{med}$), and NKT17 (PLZF$^{med}$Rorγt$^{hi}$) cells in the thymus (upper) and among total iNKT cells (bottom) are shown. **f** The CD5 expression on each stage of NKT cells determined as in Fig. 2a and presented relative to that on preselected thymocytes. The mean and SEM are shown. *$p < 0.05$, **$p < 0.01$, and ***$p < 0.001$ (two-tailed unpaired $t$ test). Representative data from 7 to 12 experiments with more than 10 mice are shown (**a**). Data pooled from 7 to 12 experiments with more than 10 mice (**b**), 2–4 experiments with more than 5 mice (**c**), and 3 experiments with 3–5 mice are shown (**d**–**f**). n.s. not significant

miRNAs that reduce the PLZF expression and thus promote differentiation into mature NKT1 cells (Supplementary Fig. 8), and we named these precursors "PLZF$^{low}$ innate precursors". These stage 0b PLZF$^{low}$ innate precursors normally differentiate into NKT1 cells. However, in the absence of let-7 miRNAs, they express high PLZF and differentiate into mature NKT2 cells, even in $Cd69^{-/-}$ mice where the "normal" NKT2 precursors have prematurely left the thymus (Fig. 6b, Supplementary Fig. 7, and Supplementary Fig. 8).

**No innate-type CD8 T cell development in $Cd69^{-/-}$ mice**. One consequence of Lin28A Tg expression is the development of CD122$^{hi}$, IL-4Rα$^{hi}$, CD44$^{hi}$, and CD24$^{lo}$ innate-type CD8 T cells, whose development is known to be dependent on IL-4 produced by NKT2 cells[4,7,31]. We found similar innate-type CD8 T cell development between Lin28A Tg $Cd69^{+/+}$ and $Cd69^{-/-}$ background mice (Fig. 6c), showing that the NKT2 cells that develop from PLZF$^{low}$ innate precursors, which become PLZF$^{hi}$ in the absence of let-7 miRNAs, are functional and produce IL-4 in the thymus.

Because of their high frequencies of IL-4-producing NKT2 cells in the thymus, BALB/c mice naturally have innate-type CD8 T cells[7]. In contrast to Lin28A Tg mice, which were found to have innate-type CD8 T cells even in the absence of CD69, we observed complete impairment of innate-type CD8 T cell development in the thymus of CD69-deficient BALB/c-background mice (Fig. 6d–f). Mature CD8SP cells in $Cd69^{+/+}$ BALB/c-background mice were CD122$^{hi}$, IL-4Rα$^{hi}$, CXCR3$^{hi}$, CD44$^{hi}$, CD24$^{lo}$, and β7integrin$^{lo}$ with high levels of $Eomes$ mRNA and the ability to produce IFNγ upon phorbol myristate acetate (PMA) + ionomycin stimulation, as previously reported[7], whereas $Cd69^{-/-}$ mature CD8SP cells did not show the innate-type phenotype (Fig. 6d–f). We thus conclude that CD69 deficiency results in not only the premature egress of CD5$^{hi}$ PLZF$^{hi}$ innate precursors and aborted NKT2 differentiation but also impaired innate-type CD8 T cell development in BALB/c-background mice due to the defective development of IL-4-producing NKT2 cells in the thymus.

## Discussion

The present study significantly enhances our understanding of very-early iNKT cell development by defining two subsets of stage 0 iNKT cells: CD24$^+$CD44$^{low}$ (stage 0a) and CD24$^+$CD44$^{hi}$ (stage 0b) innate precursor cells. We showed that CD24$^+$CD44$^{hi}$ (stage 0b) innate precursors could be further subdivided into distinct PLZF$^{hi}$ and PLZF$^{lo}$ subsets with different effector fates, as PLZF$^{hi}$ innate precursors differentiate into NKT2 cells, whereas PLZF$^{lo}$ innate precursors differentiate into NKT1 cells. Importantly, PLZF$^{hi}$ innate precursors require CD69 expression to remain in the thymus and differentiate into mature NKT2 cells, while NKT1 cells develop independently of the CD69 expression. CD69-deficient PLZF$^{hi}$ innate precursors express S1P$_1$ and prematurely exit the thymus, aborting their differentiation into mature NKT2 cells. Consequently innate-type CD8 T cells, whose development is NKT2 cell-dependent, are completely absent in $Cd69^{-/-}$ BALB/c-background mice. This study reveals that CD69 is required for the proper development of a T cell subset in the thymus, and it does so by regulating the residency time of developing immature precursor cells in the thymus.

The molecular mechanism underlying the lineage fate decisions that dictate the differentiation of the different iNKT effector subsets is not well understood. The present study reveals that lineage fate determination during iNKT cell development occurs much earlier during differentiation in the thymus than currently thought[5,6], as stage 0b CD24$^+$ CD44$^{hi}$ innate precursor cells already consist of distinct subsets with different PLZF expression and different effector fates. We suggest that these different early innate precursor subsets are the result of different intensities or durations of TCR signaling during positive selection, with strong TCR signaling generating PLZF$^{hi}$ innate precursors that differentiate into NKT2 cells and weak TCR signaling generating PLZF$^{low}$ innate precursors that differentiate into NKT1 cells. The importance of different intensities/durations of TCR signaling for the induction of different innate precursor subsets is consistent with the findings of previous reports that mature NKT2 cells require higher intensity or more persistent TCR signaling than other iNKT subsets[7,25,32,33].

In our view, strong TCR signaling during positive selection allows early innate precursors to maintain their PLZF$^{hi}$ expression and to induce high S1P$_1$ and high CD69 expression. Importantly, the high surface expression of S1P$_1$ on innate precursor cells is prevented by high CD69 expression, forcing them to remain in the thymus and complete their differentiation into mature NKT2 cells (Supplementary Fig. 5). However, because PLZF$^{hi}$ innate precursors that exited the thymus in $Cd69^{-/-}$ mice did not differentiate into mature NKT2 cells in the periphery, differentiation of PLZF$^{hi}$ innate precursors into mature NKT2 cells may not only require high PLZF but also additional intra-thymic signals, such as intra-thymic cytokine signals. In contrast to strong TCR signaling that induces high CD69 and S1P$_1$ expression in innate precursor cells, weak TCR signaling during positive selection may induce low S1P$_1$ and low CD69 expression and allow for let-7 miRNA upregulation, which directly reduces the PLZF expression. Because they express little if any S1P$_1$, PLZF$^{lo}$ innate precursors do not prematurely exit the thymus, regardless of whether or not they express CD69 (Supplementary Fig. 5 and Supplementary Fig. 8). However, some innate precursors that may have received intense TCR signaling were nonetheless found to have left the thymus, even in CD69 wild-type mice, and we named these cells early-exiting precursors (Fig. 4b, c and Supplementary Fig. 5). By regulating their thymic residency time, we think that the TCR-dependent regulation of CD69 and S1P$_1$ expression on innate precursors greatly influences their further differentiation into mature iNKT effector subsets.

While NKT2 cell differentiation is preferentially impaired in the absence of CD69, we found that the frequencies of NKT1 cells but not NKT17 cells in the thymus were significantly increased in

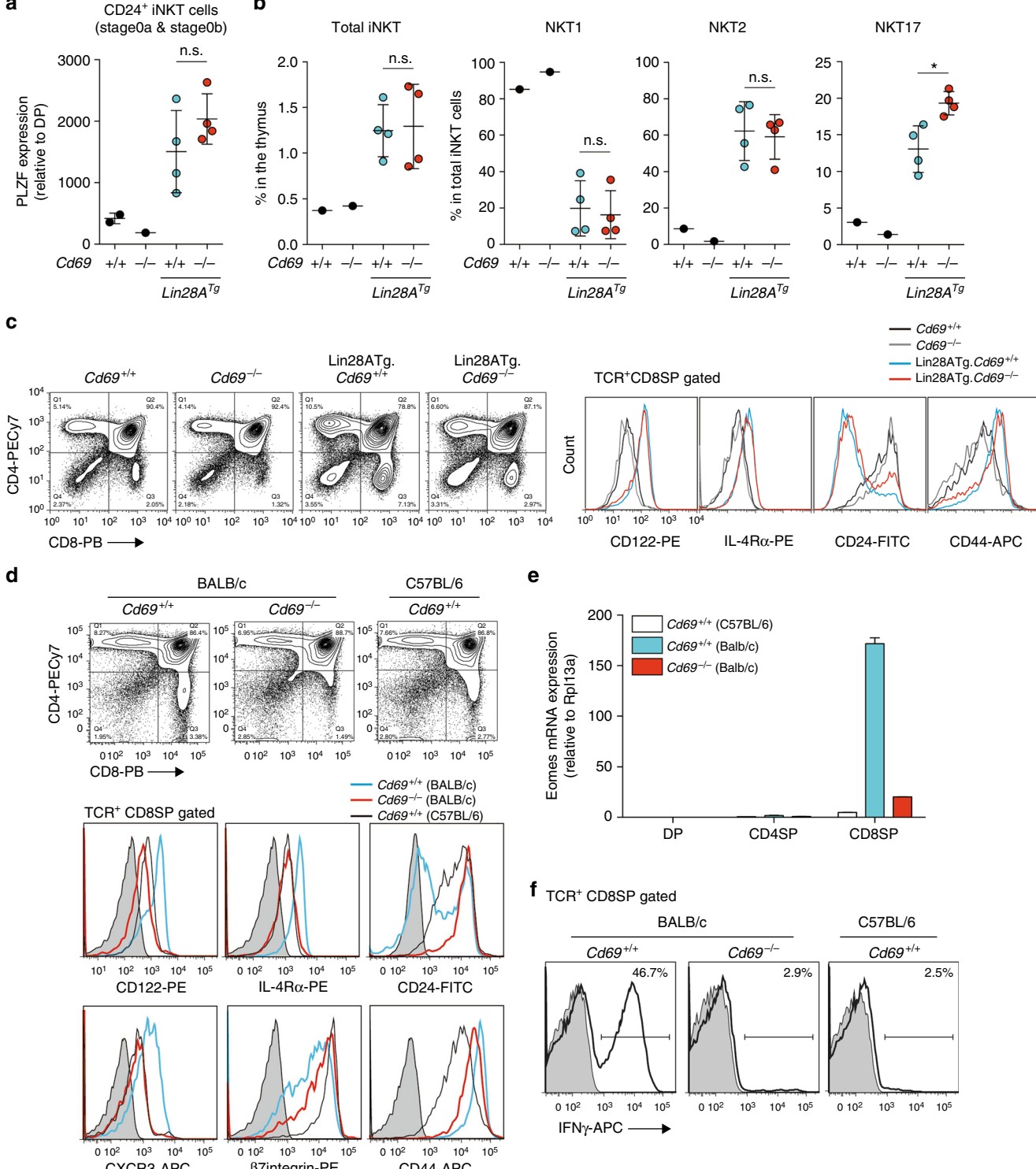

**Fig. 6** Impact of let-7 miRNA on PLZF expression in stage 0b precursors. **a** The PLZF expression in CD24+ NKT precursors (stage 0) of the indicated mice, presented relative to that in preselection thymocytes. **b** The frequencies of iNKT cells in the thymus and the frequencies of NKT1 (PLZF$^{lo}$T-bet$^+$), NKT2 (PLZF$^{hi}$Rorγt$^{med}$), and NKT17 (PLZF$^{med}$Rorγt$^{hi}$) cells among total iNKT cells of the indicated mice. **c** The CD4/CD8 profiles of the thymus from the indicated mice. The expression of CD122, IL-4Rα, CD24, and CD44 gated on TCRβ+ CD8SP cells of the indicated mice are shown. **d** The CD4/CD8 profiles of the thymus from $Cd69^{+/+}$ and $Cd69^{-/-}$ BALB/c-background mice and $Cd69^{+/+}$ C57BL/6-background mice. The expression of CD122, IL-4Rα, CD24, CXCR3, β7integrin, and CD44 gated on TCRβ+ CD8SP cells from the indicated mice are shown. **e** The *Eomes* mRNA expression relative to *Rpl13a* in sorted preselection TCRβ$^{lo-med}$ DP cells, TCRβ+CD4SP and TCRβ+CD8SP cells from the indicated mice is shown. **f** Intracellular staining of IFNγ in TCRβ+ CD8SP thymocytes from the indicated mice after PMA + ionomycin stimulation for 4 h. The mean and SEM are shown. *$p < 0.05$ (two-tailed unpaired $t$ test). Data are pooled from two experiments with four mice (Lin28A Tg) (**a**, **b**). Representative data from two experiments with more than four mice (Lin28A Tg) are shown (**c**). Representative data from more than five experiments with more than ten mice (**d**) and two experiments with three mice (**e**, **f**) are shown. n.s. not significant

BALB/c-background mice. The increased frequency of NKT1 cells in BALB/c-background mice appeared to be a compensatory response to decreased NKT2 cells. However, why only NKT1 cells are affected and not NKT17 cells is not yet clear. One possibility is that NKT1 and NKT2 cells may share a common niche, so a reduced frequency of NKT2 cells in a $Cd69^{-/-}$ thymus allows for an increased frequency of NKT1 cells. In fact, it has been reported that $Tbx21^{-/-}$ mice, which do not have NKT1 cells, have a dramatically increased number of NKT2 cells, suggesting that NKT1 and NKT2 cells do share a common niche[7].

Importantly, recent studies have reported that NKT2 cells play important roles in conditioning other developing thymocytes[7,34], as both innate-type CD8 T cell development and thymocyte emigration are regulated by NKT2 cells in the thymus[4,31,34,35]. In addition, the frequency of each iNKT subset differed significantly among mice with different backgrounds[7]. Given that the thymus produces all T cells, the difference in thymic iNKT cells may create different systemic immune responses in genetically different mice.

In this study, we showed that CD69-deficient PLZF^hi innate precursors have less residency time in the thymus than CD69-sufficient precursors and thus cannot complete their maturation into NKT2 cells (Supplementary Fig. 5). However, we also found that S1P₁ deficiency did not fully restore NKT2 cell generation in $Cd69^{-/-}$ mice or the CD5 expression on CD69-deficient CD24^+CD44^hi (stage 0b) iNKT cells. These data suggest some role of CD69 in the NKT2 cell differentiation independent of S1P₁ regulation. One possible mechanism is the existence of a CD69 ligand in the thymus that facilitates efficient TCR signaling. In fact, recent studies have shown that there are three CD69-interacting proteins[11,36–38]. We reported that myosin light chain 9/12 (Myl9/12) is the ligand for CD69 in inflammatory settings[38], and others have reported that S100A8/A9 and Galectin-1 can also interact with the CD69 molecule[36,37]. Another possibility is that CD69-mediated signaling by itself may be important for NKT2 cell differentiation. The cytosolic tail of the murine CD69 molecule contains eight serine and one threonine residue, suggesting that the CD69 molecule has signaling capability[11]. In fact, old studies have shown that the agonistic stimulation of CD69 in the presence of phorbol esters prolongs the $Ca^{2+}$ influx and extracellular signal-regulated kinase 1/2 activation in human T cells[39,40]. These data suggest possible roles of CD69-mediated signaling in NKT2 cell differentiation by prolonging TCR signaling.

In summary, we herein show that the CD69 expression on CD24^+ PLZF^hi innate precursors is essential for mature NKT2 cell differentiation in the thymus. Our data demonstrate that the CD69 expression on CD24^+ PLZF^hi innate precursors is crucial for keeping them in the thymus until they are fully differentiated into NKT2 cells. These data indicate that CD69 is required for proper thymocyte development by regulating the residency time of cells in the thymus. This insight into an important mechanism for iNKT cell development may also provide insights into regulatory mechanisms affecting the development of other T cell subsets in the thymus.

## Methods

**Mice**. C57BL/6 and BALB/c mice were obtained from CREA Inc. $Cd69^{-/-}$ [15] and $J\alpha18^{-/-}$ [41] mice were established in our laboratory and backcrossed into either BALB/c or C57BL/6 background more that 10 times. Lck proximal promoter-driven $Cd69$ Tg mice were established in our laboratory[28]. $S1pr1^{flox/flox}$ mice and CD4 promoter-driven Cre Tg mice were obtained from the Jackson Laboratory. Lin28A Tg mice were provided by Dr. A. Singer[29]. $Rag2^{GFP}$ Tg mice were provided by Dr. M. Nussenzweig[42]. Animal experiments were approved by the Chiba University Animal Care and Use Committee. Unless otherwise indicated, age-matched mice (6–14 weeks) were used for experiments. All mice were maintained under specific-pathogen-free conditions in accordance with Chiba University guidelines.

**Flow cytometry**. Single-cell suspensions were prepared and stained with fluorochrome-conjugated antibodies with the following specificities: CD3 (145-2C11), CD4 (GK1.5, RM4-5), CD5 (53-7.3), CD8α (53-6-7), CD24 (M1/69), CD25 (PC61), CD44 (IM7), CD45.1 (A20), CD45.2 (104), CD69 (H1.2F3), CD94 (18d3), CD122 (TM-β1), CD138 (281–2), β7integrin (M293), CCR7 (4B12), CXCR3 (CXCR3-173), IFNγ (XMG1.2), IL-4Rα (I015F8), NKG2D (CX5), TCRβ (H57-597), TCRγδ (GL3), PLZF (R17-809, Mags.21F7), Rorγt (Q31-378), and T-bet (Q4-46). Phycoerythrin (PE)-labeled or Alexa Fluor 647-labeled CD1d tetramers loaded with or without PBS57 were provided by the Tetramer Core Facility of the US National Institutes of Health. For cell-surface staining of fresh cells, $2 \times 10^6$ (in general) or $10 \times 10^6$ cells (for small populations, such as stage 0 cells) were incubated with 2.4G2 (anti-mouse $F_C\gamma$ III/II receptor) and stained with fluorochrome-conjugated antibodies. Dead cells were excluded by forward light-scatter gating and propidium iodide staining. For intracellular staining, live cells were first stained for cell-surface molecules and then fixed and permeabilized with the Foxp3 Transcription Factor staining buffer set (eBioscience), followed by staining for intracellular molecules. Stained samples were analyzed with a FACSLSRII or FACSCantoII flow cytometer (Becton Dickinson). Data were analyzed using the FlowJo software.

**Cell sorting**. To sort each NKT subset (i.e., NKT1, NKT2 and NKT17 cells), CD24^+ thymocytes were first removed by an autoMACS Pro Separator (Miltenyi Biotec), and then CD24^– cells were further sorted by FACSAriaIII as follows: NKT1 cells (CD3^+ CD1d.PBS57^low CD138^– CD44^hi); NKT2 cells (CD3^+ CD1d.PBS57^hi CD138^–); NKT17 cells (CD3^+ CD1d.PBS57^+ CD138^+) (Supplementary Fig. 1a). To sort CD24^+ iNKT cells, whole thymocytes were first stained with CD1d.PBS57-PE and enriched by an autoMACS Pro Separator, and then CD24^+ CD1d.PBS57^+ cells were further sorted by FACSAriaIII (Supplementary Fig. 4a). To sort DP, CD4SP, and CD8SP cells, whole thymocytes were stained with CD4-Alexa Fluor 647, CD8-Pacific Blue and H57-PE, and then sorted by FACSAriaIII as follows: DP cells (CD4^+CD8^+TCR^lo-med), CD4SP cells (CD4^+CD8^–TCR^hi), and CD8SP cells (CD4^–CD8^+TCR^hi) (Supplementary Fig. 4b).

**Mixed BM chimeras**. BM cells from $Cd69^{+/+}$ (BALB/c), $Cd69^{-/-}$ (BALB/c), and $J\alpha18^{-/-}$ (BALB/c) mice were treated with anti-Thy1 antibody, and Thy1^+ cells were depleted by an autoMACS Pro Separator (Miltenyi Biotec). A total of 12 million Thy1-depleted BM cells (2:1, 3:1, 10:1, or 14:1 mixture ratio of $Cd69^{+/+}$ or $Cd69^{-/-}$ BM cells to $J\alpha18^{-/-}$ BM cells) were injected into $J\alpha18^{-/-}$ host mice that had been lethally irradiated (4.75 Gy two times, about 12 h apart) at least 6 h before the transfer, and reconstituted mice were analyzed 7–9 weeks later.

**FTY720 injection**. FTY720 (20 μg per injection) in saline was intraperitoneally injected once a day for 5 days. The mice were analyzed 2 days after the last injection.

**In vitro stimulation of thymocytes**. Whole thymocytes were stimulated with PMA (1 ng ml^−1) and ionomycin (500 nM) for 4 h in the presence of BD GolgiStop (BD Biosciences). Stimulated cells were harvested, intracellular staining was performed, and cells were analyzed by flow cytometry.

**Quantitative RT-PCR**. Total RNA was isolated with Trizol and treated with DNase I (Invitrogen) to eliminate possible genomic DNA contamination. Complementary DNA synthesis was performed using a Superscript III First-Strand Synthesis System for reverse transcription-polymerase chain reaction (RT-PCR) Kit (Invitrogen) with oligo dT primers. Quantitative RT-PCR was performed with an ABI PRISM 7500 Sequence Detection System. All mRNA values are shown relative to $Rpl13a$. The probes for the detection of genes in this study were purchased from Roche Applied Science. Probes and primers are listed in Supplementary Table 1.

**Statistical analyses**. The sample size was determined empirically, and at least two independent experiments were conducted. Animals were allocated to groups based on their genotype and no blinding was used. Intergroup variance was assessed using the $F$ test. For normally distributed data, statistical significance was determined using a paired or unpaired two-sided Student's $t$ test; an unpaired two-sided Welch's $t$ test was used to analyze groups that showed differing variance. For the multiple comparison, a one-way analysis of variance was used for the statistical analysis. $P$ values of 0.05 or less were considered significant.

## Data availability

All data that support the findings of this study are available from the corresponding author upon reasonable request.

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

## Acknowledgements

We thank Drs. Ruth Etzensperger and Belgacem Mihi for their critical reading of the manuscript; M. Nussenzweig (Rockefeller University) for *Rag2*^GFP Tg mice; the NIH Tetramer facility for CD1d tetramer PBS57 reagents; and K. Sugaya for their expert performance of flow cytometry. This research was supported by the Ministry of Education, Culture, Sports, Science, and Technology, Japan; Grant-in-Aid for Scientific Research (C) (15K08524) and a Grant-in-Aid for Scientific Research on Innovative Areas (Research in a proposed research area) (17H05787); The Hamaguchi Foundation for the Advancement of Biochemistry; Astellas Foundation for Research on Metabolic Disorders; The Uehara Memorial Foundation.

## Author contributions

M.Y.K. designed the study, performed experiments, analyzed data, and wrote the manuscript. A.I., K.H., Y.M., M.S., T.K., Y.E., S.O., R.Y., and S.M. performed experiments and analyzed data. A.S. provided the mice. A.S. and T.N. provided helpful discussions and contributed to writing the manuscript.

## Additional information

**Competing interests:** The authors declare no competing interests.

