## [Peer Review File · Nature Communications]

Reviewers' comments:

Reviewer #1 (NKT, CD1d)(Remarks to the Author):

This paper has investigated the role of CD69 in the development of iNKT cells. It is found that CD69 expression in the thymus is critically important for the development of NKT2 cells but not NKT1 or NKT17 cells. Since NKT2 cells do not express CD69, precursors were investigated, which identified a CD69-expressing precursor to NKT2 cells that expresses high levels of the innate transcription factor PLZF, and shows signs of strong TCR signaling. In CD69-deficient mice, such precursors were able to prematurely exit the thymus, thus preventing the generation of NKT2 cells, and conversely, in CD69 transgenic mice, NKT2 cells were enriched. Mechanistically, CD69 is an inhibitor of the S1P1 receptor that is critical for the exit of thymocytes from the thymus. Consistent with this notion, it was shown that S1P1 receptor knockouts have significantly increased levels of NKT2 cells, and S1P1 receptor deficiency or an S1P1 inhibitor were able to rescue NKT2 cell development in CD69-deficient mice. Finally, it is also shown that CD69-deficient mice have a defect in the generation of innate-type CD8 T cells, which were previously shown to require IL-4 production by thymic NKT2 cells for their development. From these findings, the authors conclude a critical role for CD69 in the thymic retention of a precursor to the development of NKT2 cells.

Specific comments: This is a nice paper that identifies a unique precursor to the NKT2 cell lineage in the thymus and further shows a critical role for CD69 in the retention of this precursor in the thymus to permit the development of NKT2 cells. The experiments are performed with a variety of experimental models and are convincing. As distinct iNKT cell subsets exhibit diverse therapeutic properties, the findings are relevant to treatment of human disease.

Reviewer #2 (Thymocyte development, ILC)(Remarks to the Author):

NKT cells, like innate lymphocytes, adopt effector programs during their development in the thymus. The authors find the CD69 deficiency impacts the frequency of NKT1 and NKT2 cells in the thymus. They reveal a mechanism whereby strong TCR signals upregulate CD69 preferentially on NKT cell precursor cells, and CD69 prevents thymic egress via its known effects on S1PR1 function. In the absence of CD69, precursors of NKT2 cells appear to egress the thymus prematurely, preventing NKT2 cell maturation. Importantly, NKT2 numbers remained low in spleen, confirming that CD69 deficiency reduced generation of NKT2 cells. In Balb/c mice, that have high frequencies of NKT2 cells that make IL-4, innate CD8 T cells develop; the generation of these innate cells is also impacted by CD69 deficiency. The authors link TCR signaling intensity (via examining CD5 and Egr1 levels that are regulated by TCR signals) on NKT2 cell precursors as important for CD69 upregulation, and further suggest that the let7 miRNAs might regulate NKT2 development via effects on PLZF.

There are many interesting observations in this paper, and the authors have refined our understanding of precursors of NKT cells. However, there are also some defects that should be remedied. Most importantly, effects of CD69 deficiency are not shown to be cell intrinsic. It is possible that CD69 deficiency allows the premature egress of other cells that are important for the generation of NKT2 cells. The results should be confirmed in mixed stem-cell chimeras made with wild-type and CD69 deficient bone marrow to establish that CD69 is acting in a cell-autonomous fashion, as the authors propose.

Additional points:

In general, histograms of expression should be shown in addition to the statistics, to clarify how

homogenous are the populations, and what are the gates used to define positive cells.

Ex: NK Receptors and CD69 in Fig 1 b and c, GFP in Fig 2a, NK Receptors and CD69 in Fig 2 b and c,...

The plots on Figure 3a showing % of NKT1/2/3 are labelled incorrectly.

Although CD69 deficiency is not reported to impact thymic cellularity, the authors should establish that this is the case in their colony, for both B6 and Balb/c mice, to provide confidence that presentation of percentages rather than absolute cell numbers. Certainly in Figure 3, absolute numbers of total NKT and NKT1/2/3 should be indicated instead or in addition to relative % to show which population(s) drives the changes.

Reviewer #1 (NKT, CD1d)

Remarks to the Author:

This paper has investigated the role of CD69 in the development of iNKT cells. It is found that CD69 expression in the thymus is critically important for the development of NKT2 cells but not NKT1 or NKT17 cells. Since NKT2 cells do not express CD69, precursors were investigated, which identified a CD69-expressing precursor to NKT2 cells that expresses high levels of the innate transcription factor PLZF, and shows signs of strong TCR signaling. In CD69-deficient mice, such precursors were able to prematurely exit the thymus, thus preventing the generation of NKT2 cells, and conversely, in CD69 transgenic mice, NKT2 cells were enriched. Mechanistically, CD69 is an inhibitor of the S1P1 receptor that is critical for the exit of thymocytes from the thymus. Consistent with this notion, it was shown that S1P1 receptor knockouts have significantly increased levels of NKT2 cells, and S1P1 receptor deficiency or an S1P1 inhibitor were able to rescue NKT2 cell development in CD69-deficient mice. Finally, it is also shown that CD69-deficient mice have a defect in the generation of innate-type CD8 T cells, which were previously shown to require IL-4 production by thymic NKT2 cells for their development. From these findings, the authors conclude a critical role for CD69 in the thymic retention of a precursor to the development of NKT2 cells.

Specific comments: This is a nice paper that identifies a unique precursor to the NKT2 cell lineage in the thymus and further shows a critical role for CD69 in the retention of this precursor in the thymus to permit the development of NKT2 cells. The experiments are performed with a variety of experimental models and are convincing. As distinct iNKT cell subsets exhibit diverse therapeutic properties, the findings are relevant to treatment of human disease.

Our response:

Thank you very much for your positive comments, which encourage us very much.

Reviewer #2 (Thymocyte development, ILC)

Remarks to the Author:

NKT cells, like innate lymphocytes, adopt effector programs during their development in the thymus. The authors find the CD69 deficiency impacts the frequency of NKT1 and NKT2 cells in the thymus. They reveal a mechanism whereby strong TCR signals upregulate CD69 preferentially on NKT cell precursor cells, and CD69 prevents thymic egress via its known effects on S1PR1 function. In the absence of CD69, precursors of NKT2 cells appear to egress the thymus prematurely, preventing NKT2 cell maturation. Importantly, NKT2 numbers remained low in spleen, confirming that CD69 deficiency reduced generation of NKT2 cells. In Balb/c mice, that have high frequencies of NKT2 cells that make IL-4, innate CD8 T cells develop; the generation of these innate cells is also impacted by CD69 deficiency. The authors link TCR signaling intensity (via examining CD5 and Egr1 levels that are regulated by TCR signals) on NKT2 cell precursors as important for CD69 upregulation, and further suggest that the let7 miRNAs might regulate NKT2 development via effects on PLZF.

There are many interesting observations in this paper, and the authors have refined our understanding of precursors of NKT cells. However, there are also some defects that should be remedied. Most importantly, effects of CD69 deficiency are not shown to be cell intrinsic. It is possible that CD69 deficiency allows the premature egress of other cells that are important for the generation of NKT2 cells. The results should be confirmed in mixed stem-cell chimeras made with wild-type and CD69 deficient bone marrow to establish that CD69 is acting in a cell-autonomous fashion, as the authors propose.

Our response:

Thank you very much for your valuable suggestions. As requested, we performed the mixed bone marrow chimera experiment. While preparing to construct BM chimera mice, we found that some iNKT cells lose Thy1 expression so that Thy1 cannot be used as a congenic marker in mixed radiation bone marrow chimeras; therefore, we could not do the simple mixed BM ($Cd69^{+/+}$ + $Cd69^{-/-}$ BMs) chimera experiment (Our lab

only has *Thy1* congenic mouse in BALB/c background). In addition, we also found that some thymic iNKT cells are radio-resistant so that we could not use normal BALB/c as irradiated host mice. Accordingly, we set up the chimeras in which *Cd69*^{+/+} or *Cd69*^{-/-} BM cells mixed together with *Ja18*^{-/-} BM cells in various ratio were injected into *Ja18*^{-/-} host mice. In these chimera mice, all iNKT cells were derived from either *Cd69*^{+/+} or *Cd69*^{-/-} BM cells and developed in the environment where many cells were CD69-sufficient. We found the specific reduction of NKT2 cells in the absence of CD69 in these chimera mice regardless of mixture ratio of BM cells (Supplementary figure 3e). This result demonstrates that the effect of CD69 deficiency on NKT2 cells is cell-intrinsic. We have included the new data in new Supplementary figure 3e and added the description in the text (highlighted in red characters) in p8 line 17~ p9 line 2.

Additional points:

In general, histograms of expression should be shown in addition to the statistics, to clarify how homogenous are the populations, and what are the gates used to define positive cells. Ex: NK Receptors and CD69 in Fig 1 b and c, GFP in Fig 2a, NK Receptors and CD69 in Fig 2 b and c...

Our response:

Thank you very much for your comments. As requested, we added the histograms related to Fig1b, 1c, 2a, 2b and 2c in Supplementary figures S1d, S1e, S2a, S2c and S2d, respectively.

The plots on Figure 3a showing % of NKT1/2/3 are labeled incorrectly.

Our response:

Thank you very much for your comments. We re-labeled Figure 3a.

Although CD69 deficiency is not reported to impact thymic cellularity, the authors should establish that this is the case in their colony, for both B6 and Balb/c mice, to provide confidence that presentation of percentages rather than absolute cell numbers. Certainly in Figure 3, absolute numbers of total NKT and NKT1/2/3 should be indicated instead or in addition to relative % to show which population(s) drives the changes.

Our response:

Thank you very much for your comments. As a reviewer expected, thymic cellularity in Cd69^{-/-} mice was similar to Cd69^{+/+} mice in the both cases of B6 and BLAB/c background mice in our colony. To provide confidence, we add the data of absolute numbers of total NKT and NKT1, 2, 17 cells from both B6 and BALB/c background mice in the thymus and periphery in new supplementary figure S3a, S3c and S3d. The data showed the confidential result that NKT2 cells were significantly diminished in the absence of CD69.

REVIEWERS' COMMENTS:

Reviewer #2 (Remarks to the Author):

This is well revised and I am supportive.